# Inhibition of EIF2α Dephosphorylation Decreases Cell Viability and Synergizes with Standard-of-Care Chemotherapeutics in Head and Neck Squamous Cell Carcinoma

**DOI:** 10.3390/cancers15225350

**Published:** 2023-11-09

**Authors:** Anna M. Cyran, Florian Kleinegger, Norbert Nass, Michael Naumann, Johannes Haybaeck, Christoph Arens

**Affiliations:** 1Legorreta Cancer Center, Department of Pathology and Laboratory Medicine, Brown University, Providence, RI 02906, USA; 2Department of Otorhinolaryngology, Head and Neck Surgery, Otto-von-Guericke University, 39120 Magdeburg, Germany; 3Diagnostic & Research Center for Molecular Biomedicine, Institute of Pathology, Medical University of Graz, 8010 Graz, Austriajohannes.haybaeck@tyrolpath.at (J.H.); 4Institute of Pathology, University Hospital Brandenburg, Brandenburg Medical School Theodor Fontane, 14770 Brandenburg an der Havel, Germany; norbert.nass@mhb-fontane.de; 5Institute of Experimental Internal Medicine, Otto von Guericke University, 39120 Magdeburg, Germany; naumann@med.ovgu.de; 6Department of Otorhinolaryngology, Head and Neck Surgery, Giessen and Marburg University Hospitals, Campus Giessen, 35392 Giessen, Germany; christoph.arens@hno.med.uni-giessen.de

**Keywords:** head and neck cancer, HPV, dysplasia, salubrinal, translation initiation, EIF2alpha, drug synergy, cisplatin, bortezomib

## Abstract

**Simple Summary:**

Head and neck squamous cell carcinoma (HNSCC) arises in the mucosal membranes in the head and neck and constitutes 3.9% of new cancer cases annually. Its pathogenesis is stepwise; the formation of invasive and metastatic cancer is preceded by premalignant mucosal changes. With little improvement in overall survival over the last decade, HNSCC remains an important cause of patient morbidity. Therapy options include surgery, radiochemotherapy, and immunotherapy, but resistance is common. A way to overcome drug resistance is to target cellular processes present in all cells, such as protein biosynthesis, i.e., translation. This study found that eukaryotic translation initiation factor 2α (EIF2α) is highly expressed in HNSCC in comparison to normal tissue, and its expression mirrors the progression to malignancy. Pharmacological targeting of EIF2α by inhibition of its dephosphorylation with salubrinal decreases cell viability, the ability to form colonies, and disrupts the cell cycle. Salubrinal also works synergistically with standard chemotherapeutics.

**Abstract:**

Drug resistance is a common cause of therapy failure in head and neck squamous cell carcinoma (HNSCC). One approach to tackling it is by targeting fundamental cellular processes, such as translation. The eukaryotic translation initiation factor 2α (EIF2α) is a key player in canonical translation initiation and integrates diverse stress signals; when phosphorylated, it curbs global protein synthesis. This study evaluates EIF2α expression and phosphorylation in HNSCC. A small-molecule inhibitor of EIF2α dephosphorylation, salubrinal, was tested in vitro, followed by viability assays, flow cytometry, and immunoblot analyses. Patient-derived 3D tumor spheres (PD3DS) were cultured with salubrinal and their viability assessed. Lastly, salubrinal was evaluated with standard-of-care chemotherapeutics. Our analysis of RNA and proteomics data shows elevated EIF2α expression in HNSCC. Immunohistochemical staining reveals increasing EIF2α abundance from premalignant lesions to invasive and metastatic carcinoma. In immunoblots from intraoperative samples, EIF2α expression and steady-state phosphorylation are higher in HNSCC than in neighboring normal tissue. Inhibition of EIF2α dephosphorylation decreases HNSCC cell viability and clonogenic survival and impairs the G_1_/S transition. Salubrinal also decreases the viability of PD3DS and acts synergistically with cisplatin, 5-fluorouracil, bleomycin, and proteasome inhibitors. Our results indicate that pharmacological inhibition of EIF2α dephosphorylation is a potential therapeutic strategy for HNSCC.

## 1. Introduction

Head and neck cancers are a heterogeneous group of malignancies, accounting annually for over 750,000 new cases and 360,000 deaths worldwide. In Europe, around 150,000 new cases are diagnosed, leading to 67,000 deaths a year [1]. Ninety-one percent of head and neck cancers are squamous cell carcinomas (HNSCC) arising from the mucosal lining of the upper aerodigestive tract [2]. Despite the decline in alcohol and tobacco use as the most common risk factors for the disease, the incidence of laryngeal and hypopharyngeal cancers remains stagnant, while the incidence of cancers in the oral cavity and oropharynx, associated with oncogenic human papillomaviruses (HPV), is on the rise [3]. The prognosis depends on several factors, including anatomical location and HPV status. The larynx is most frequently affected and is associated with the best 5-year overall survival (OS) rates, 59% Europe-wide. The 5-year OS for other locations varies between 25–46%, with little improvement over the last decade [2]. HPV-positive HNSCC, though often more locally advanced at diagnosis, is associated with a better prognosis [4]. HNSCC is associated with a unique disease-specific morbidity, including airway obstruction, loss of speech, and dysphagia, and significantly compromises patients’ quality of life. Multimodal therapy is the cornerstone of treatment, including cold steel surgery, conventional radio-chemotherapy, and recently introduced immunotherapy. Nevertheless, constitutive and acquired resistance remain major obstacles, and few therapeutic options exist for patients receiving second-line systemic therapy, prompting the search for new therapeutic approaches [3,5,6,7].

The pathogenesis of HNSCC is stepwise, starting with epithelial dysplasia, progressing to carcinoma in situ, and finally invasive cancer. Some early genetic alterations include 3p, 9p, and 17p loss of heterozygosity, as well as cyclin-dependent kinase inhibitor 2A (CDKN2A) and tumor protein 53 (TP53) inactivation seen already in dysplasia. In carcinoma, 11q, 4q, and 8p alterations, cyclin D1 (CCND1) amplification, and phosphatase and tensin homolog (PTEN) inactivation are frequent [8]. Overall, HNSCC is very heterogeneous in its genetic landscape, and mutations in tumor suppressor genes are more common than in oncogenes, making the implementation of targeted therapies more challenging [9]. Targeting other cellular pathways on which cancers depend for survival is an alternative to oncogene-directed therapies, which only apply in selected cases. As these pathways are also present in normal cells, this phenomenon was dubbed non-oncogene addiction [10,11]. An example of such an approach is targeting protein synthesis. Translation is a fundamental process tightly regulated during its initiation phase. It plays a crucial role in gene expression. Its dysregulation is a hallmark of cancer and leads to abnormal proliferation, angiogenesis, metabolic changes, and patient survival [12]. There is ample evidence for altered expression of eukaryotic translation initiation factors (eIF) in human cancers. Robust proliferation in cancer cells leads to an increased demand for protein synthesis, suggesting there may be a therapeutic window for pharmacologic interventions [13]. In fact, modulating translation through pharmacological means has a favorable safety profile for several other cancer types [12].

Eukaryotic translation begins with the formation of the ternary complex consisting of the initiator Met-tRNAi, guanosine triphosphate (GTP), and the heterotrimeric EIF2. In cooperation with EIF3, EIF1, eIF1A, and EIF5, the ternary complex then binds to the ribosomal 40S subunit, forming the 43S preinitiation complex necessary for the translation of most cellular mRNA [14]. Tumor cells endure elevated proteotoxic, mitotic, metabolic, and oxidative stress levels. A critical regulatory node integrating cells’ homeostatic needs and environmental stimuli is the alpha subunit of the eukaryotic initiation factor 2 (EIF2α). It is regulated by phosphorylation by four kinases, each responding to a specific type of stress. The protein kinase R (PKR) responds to dsRNA in viral infections; PKR-like ER kinase (PERK) acts to counter endoplasmic reticulum (ER) and oxidative stress; general control nonderepressible 2 (GCN2) senses nutrient depletion; and heme-regulated EIF2α kinase (HRI) senses heme deprivation, drugs, and toxicants [15,16]. Phosphorylation of EIF2α at serine 51 initiates the integrated stress response (ISR) and stalls global translation. This crucial regulatory mechanism allows time for repair following diverse insults before cells continue to proliferate [17]. At the same time, ISR promotes the translation of mRNA harboring upstream open reading frames (uORF) and internal ribosome entry sites (IRES) in the 5′ untranslated region (UTR) [18], most prominently the activating transcription factors (ATFs) and DNA-damage inducible transcript 3 (DDIT3, CHOP) [19]. These genes serve as adaptations to stress and are often inefficiently translated in normal conditions. The ISR is designed to counteract noxious stimuli rapidly. Once cellular homeostasis is restored, the signal is extinguished, and EIF2α is dephosphorylated by growth arrest and DNA-damage inducible protein 34 (GADD34), which is preferentially translated during stress and recovery [20].

The effects of EIF2α phosphorylation may be cytoprotective or deleterious, depending on the context: the type of environmental cue, its duration, and concomitant signaling. Extensive data show that prolonged activation of this pathway can lead to programmed cell death or growth arrest [20,21]. Here, we describe the expression pattern of *EIF2S1*, which encodes the regulatory subunit of EIF2α in HNSCC. We determined the protein abundance in carcinoma and premalignant epithelial lesions and related the expression to clinical and histopathological traits. We then evaluated the effects of pharmacological targeting of EIF2α by a small-molecule inhibitor to determine its therapeutic potential in HNSCC alone and in combination with standard-of-care drugs.

## 2. Materials and Methods

### 2.1. Collection of Samples and Ethics Statement

The study was conducted in accordance with the Declaration of Helsinki and received the necessary Ethics Committee approval (114/17, approved on 20 July 2017). The retrospective arm of the study utilizes histological slides from 83 patients diagnosed with HNSCC at the Otto-von-Guericke University in Magdeburg, Germany, and the Medical University of Graz, Austria. Sixteen patients undergoing tumor resection at the Otto-von-Guericke University in Magdeburg consented to the intraoperative collection of tissue fragments, which were unnecessary for diagnostic histopathology. The harvested tumor fragments were frozen in liquid nitrogen and used for protein isolation. All samples have been pseudonymized before processing for research purposes.

### 2.2. Bioinformatic Analysis of EIF2S1 and EIF2α Expression

mRNA expression data provided by the Cancer Genome Atlas initiative (TCGA) [22] was accessed and analyzed via the UALCAN web platform [23,24]. The human papillomavirus (HPV) status used for this analysis was determined in a separate study using DNA sequencing and PathSeq methods [25]. Gene expression alterations were analyzed via cBioPortal [9,26] using the HNSCC TCGA, Firehose Legacy dataset. Kaplan-Meier survival analysis of TCGA data was performed via the GEPIA [27] web server. Patients were divided into high- and low-expressors at median; statistical significance is reported as a log-rank test *p*-value. The Clinical Proteomic Tumor Analysis Consortium (CPTAC) [28] protein expression data were analyzed via the UALCAN web platform. Protein expression is expressed as Z-values, representing the number of standard deviations (SD) from median across samples in each cancer entity. Z-values are calculated from Log_2_ spectral count ratios obtained via CPTAC, normalized to the sample profile, and across the dataset.

### 2.3. Immunohistochemistry (IHC)

Tissue specimens were fixed in formalin, embedded in paraffin, and stained with hematoxylin and eosin (H&E) according to standard methods. Tissue block sections (4 μm) were mounted on adhesive-coated glass slides. IHC was performed automatically using a Ventana Immunostainer XT and UltraView Universal DAB detection kit (both Ventana Medical Systems, Tucson, AZ, USA). Briefly, the procedure included deparaffinization, 30-min conditioning, heat-induced epitope retrieval (HIER), primary antibody incubation for 30 min, and peroxidase-labeled secondary antibody incubation. Information on antibodies used in this study is provided in Appendix A. Two pathologists (S.S. and P.C.) independently scored slides using light microscopy. Staining was evaluated using an intensity score (IS; 0 = no staining; 1 = weak staining; 2 = moderate staining; and 3 = strong staining) and a proportion score defined as: <20% of cells = 1; 21–50% = 2; 51–80% = 3; >80% = 4. (PS). The total immunostaining (TIS) score was calculated by multiplying IS by PS. Infection with HPV was determined by immunostaining against P16 protein and HPV 16/18 early antigen 6. Secondary reagents used for P16 and HPV16/18 evaluation: BIOLOGO Universal Staining System DAB (Art. No. DA005, Exalpha Biologicals Inc., Shirley, MA, USA).

### 2.4. Immunoblotting

Intraoperative specimens were immediately frozen in liquid nitrogen and stored at −80 °C. Cryosections were prepared to confirm or exclude the presence of a tumor. Tissue samples were homogenized (MagNa Lyser, Roche Diagnostics, Rotkreuz, Switzerland) and incubated with NP-40 lysis buffer with a phosphatase inhibitor (#4906845001, PhosSTOP, Millipore Sigma, Darmstadt, Germany). The later steps, including protein quantification by Bradford assay (Protein Assay Dye Reagent, #500-006; Bio-Rad Laboratories Inc., Feldkirchen, Germany), were the same for tissue- and cell culture-derived lysates. 10% SDS-PAGE gels were loaded with 20 μg protein lysate per well. Following gel electrophoresis, proteins were transferred onto a PVDF membrane by semi-dry transfer. The non-specific signal was blocked with 5% non-fat milk for 1 h at room temperature. The membrane was then incubated overnight at 4 °C with primary antibodies. The next day, a secondary antibody (horseradish peroxidase-conjugated) was added for 1 h at room temperature. For visualization of proteins obtained from cryo-samples, the Image Quant Chemiluminescent Imaging System (LAS 500, Cytiva, Avantor, Radnor, PA, USA) and the Chemiluminescence Detection Reagent Kit (#WBKLS0500, MERCK, Darmstadt, Germany) were used. Visualization of immunoblots from cell culture was performed using Pierce ECL Western Blotting Substrate (#32106; ThermoFischer Scientific, Waltham, MA, USA); Blue Autoradiography and Western Blotting Film (#*1968-3810*; USA Scientific, Ocala, FL, USA); and Agfa CP1000 Developer (Agfa, Mortsel, Belgium).

### 2.5. Cell Culture and Preparation of Stock Solutions

Human hypopharyngeal (FaDu) and oral (SCC4) squamous cell carcinoma cells were purchased from Creative Bioarray (New York, NY, USA). Cells were cultured in EMEM (FaDu) and DMEM:F12 (SCC4) supplemented with 10% FBS and 1% P/S (all supplied by Gibco, Life Technologies, Carlsbad, CA, USA). For passaging, a 0.05% trypsin solution with EDTA (Gibco, Life Technologies, USA) was used. Cells were incubated at 37 °C in a humidified atmosphere with 5% CO_2_. Salubrinal (CAS No. 405060-95-9; Cayman Chemical, Ann Arbor, MI, USA) was dissolved in DMSO (CAS No. 67-68-5; Sigma Millipore). Stock solutions were stored at −20 °C. Further dilutions were made with the cell culture medium immediately before each experiment. Other chemicals used: cisplatin (CAS No. 15663-27-1; Sigma Millipore), 5-fluorouracil (CAS No. 51-21-8; Sigma Millipore), bleomycin (CAS No. 9041-93-4; Sigma Millipore), paclitaxel (CAS No. 33069-62-4; Sigma Millipore), bortezomib (CAS No. 179324-69-7; Sigma Millipore), and MG-132 (CAS No. 13340-82-67; Sigma Millipore).

### 2.6. Viability Testing and Calculation of Drug Synergies

Cell viability was assessed by colorimetric measurement of the conversion of 3-(4,5-dimethylthiazol-2-yl)-2,5-diphenyltetrazolium bromide (MTT, #M6494, ThermoFischer Scientific, USA) to formazan. Cells were seeded onto a 96-well plate (2000 cells/well). The next day, the desired concentrations of tested chemicals and solvent controls were added. Cells were treated for 24, 48, and 72 h. At each timepoint, 10 μL of MTT solution in phosphate-buffered saline (PBS) was added per 100 μL of total well volume. Plates were incubated at 37 °C for 2 h. The supernatant was discarded, and 100 μL of DMSO was added to each well. Following a 15-min incubation on a shaker, optical density was measured at 562 nm with the GloMax^®^-Multi Detection System (Promega, Madison, WI, USA). Each experiment was performed in triplicate.

Data from viability assays for drug combinations were analyzed via a web-based platform (SynergyFinder.fimm.fi, (accessed on 9 August 2023) [29]. A 4-parameter logistic regression algorithm was used for single-agent curve fitting and generating dose-response matrices. The observed response to drug combinations was compared to the predicted response, assuming no interaction between the two substances. The reference null model used is the highest single agent (HSA) [29].

### 2.7. Colony Formation Assay

Cells were seeded onto a 6-well plate, 200 cells per plate. The next day, the desired concentrations of tested chemicals were added. On day 10 of the experiment, cells were washed with PBS, fixed in methanol, and stained for 1 h in a 1:10 Giemsa solution (Dr. K. Hollborn and Soehne, Leipzig, Germany). Plates were washed with distilled water, and colonies formed were counted manually.

### 2.8. Cell Death and Cell Cycle Analysis with Flow Cytometry

Asynchronous cells were harvested by trypsinization and re-suspended in ice-cold PBS. Seventy percent ethanol was gradually added while vortexing for overnight fixation at 4 °C. The next day, cells were washed twice, suspended in 1 mL of PBS, and incubated with 50 μL of RNase (working solution 100 μg/mL, Cat. No. 19101, Qiagen, Venlo, The Netherlands) to ensure exclusive staining of DNA. Next, 50 μL propidium iodide (50 μg/mL; #P4864, Sigma-Aldrich, St. Louis, MO, USA) was added, staining for 30 min at room temperature. Percentages of cells in each phase of the cell cycle were determined with a FACS-Calibur flow cytometer (BD Biosciences, San Jose, CA, USA) and used for statistical analysis. 

### 2.9. Chemosensitivity Testing of HNSCC Patient-Derived 3D Tumor Spheres

The protocol used was based on previously published work by Boehnke et al. [30]. Mechanical and enzymatic tissue dissociation was adjusted every time to individual tissue characteristics. Cryosections of intraoperative tissue samples were performed to confirm the presence of a tumor. Tumor fragments were immersed in complete medium (EMEM, 10% FBS, 1% P/S) with the addition of 2.0 μg/mL amphotericin B (E3789-1g, Sigma Millipore, Germany) overnight. The following day, upon receipt in the laboratory, the tissue was mechanically fragmented with sterile scalpels (Cat. No. 22-079-707, ThermoFischer Scientific, USA), enzymatically digested with collagenase IV (C4-BIOC, Sigma-Aldrich, Darmstadt, Germany), DNase I solution (#07900, Stemcell Technologies, Kent, WA, USA), and dispase solution 5 U/mL (#07913, Stemcell Technologies, USA). The resulting suspension was passed through a 70 μM pore-size strainer (#27216, Stemcell Technologies, USA). The residue was discarded, and the aggregates in the flow-through were passed through a 37 μM strainer (#27250, Stemcell Technologies, USA). This time, the flow-through was discarded, while residual aggregates were washed off with complete medium into a test tube and centrifuged. The obtained cells were suspended in supplemented medium, seeded as an adherent monolayer in 12-well plates, and expanded for 2–3 passages at 37 °C, 5% CO_2_. Once 70–80% confluence was achieved, cells were harvested enzymatically (TrypLE Express Enzyme, #12604013, ThermoFisher Scientific, USA), centrifuged, and suspended in the supplemented medium. For chemosensitivity testing, 5000 cells per well suspended in a 1:2 mixture of Matrigel and medium were seeded onto a 384-well plate with an automated liquid handler (Biomek FX P Liquid Handler, Beckman Coulter, Brea, CA, USA). On day 4, test substances were added. Salubrinal concentrations used: 0.4, 2, 20, and 50 μM.

After 4 days of incubation, an ATP-based viability assay was performed according to the manufacturer’s instructions (CellTiterGlo, #G9241, Promega, USA). Wells containing Matrigel only were used for background subtraction. DMSO was used as a solvent for all substances and controls. Four replicates were prepared for each dose. The percentage viability was determined as follows:$$\%\;viability=100\times \frac{{Luminescence}_{READOUT}-{Luminescence}_{BLANK}}{{Luminescence}_{DMSO}-{Luminescence}_{BLANK}}$$

### 2.10. Statistical Analysis

Discrete data obtained from IHC are represented as medians with 95% confidence intervals (CI). Differences between groups were tested using the X^2^ test. Continuous data are expressed as means +/− SD. Inter-parameter correlations were determined using the non-parametric Spearman’s rho. The significance level was set to *p* < 0.05. All statistical analyses and graphs were generated using GraphPad Prism 9 (GraphPad Software Inc., La Jolla, CA, USA).

## 3. Results

### 3.1. The Eukaryotic Initiation Factor 2 (EIF2) Genes and Their Protein Products Are Overexpressed in Head and Neck Squamous Cell Carcinoma (HNSCC)

We set out to characterize the expression patterns of genes encoding members of the EIF2 heterotrimer. *EIF2S1*, encoding the α subunit, is overexpressed in HNSCC (n = 520) in comparison to normal tissue (n = 44) (*p*-value = 1.62 × 10^−12^). *EIF2S2* and *EIF2S3*, encoding subunits β and γ, are also highly expressed in HNSCC (*p* < 1 × 10^−12^ and *p* = 3.69 × 10^−14^, respectively) (Figure 1A, Appendix A). *EIF2S1* expression correlates negatively in Kaplan-Meier analysis with overall survival (OS) (log-rank test; *p* = 0.00016) and disease-free survival (DFS) (log-rank test; *p* = 0.044) (Figure 1B,C). It is also associated with clinicopathological parameters indicating disease aggressiveness and severity: extracapsular spread in nodal metastases (*p* = 0.039), perineural invasion (*p* = 0.0154), and adjuvant postoperative systemic therapy administration (*p* = 3.722 × 10^−3^). The role of EIF2 in translation depends on its interaction with EIF2B, a stoichiometrically less abundant, multi-subunit protein complex acting as a GTP-exchange factor. EIF2B comprises five subunits (α, β, γ, δ, and ε) encoded by *EIF2B1-5*. All components, except *EIF2B4*, are also overexpressed in HNSCC, often showing a progression with cancer stage and grade (Appendix A). Genes encoding EIF2 components are rarely mutated (<1%). Their elevated expression is associated with decreased promoter methylation at cancer stages 2, 3, and 4, as well as copy number amplifications in *EIF2A* and *EIF2B5* genes. An exception is *EIF2S3*, which is more commonly deleted than amplified (Figure 1D, Appendix A).

Next, we analyzed data from a mass spectrometry-based proteomics assay provided by CPTAC to characterize EIF2 complex protein abundance. In a dataset including 108 HNSCC samples and 71 controls, EIF2α is significantly overexpressed in cancer (*p*-value = 7.56 × 10^−16^) (Figure 1E). EIF2α is more elevated at advanced stages and most abundant in grade 1 tumors, followed by grade 2. Though average protein expression in grade 3 tumors was higher than in controls, the difference was not significant, likely due to the low number of undifferentiated tumors in the dataset (Appendix A). Similarly, EIF2β and EIF2γ are overexpressed in HNSCC (*p* = 1.59 × 10^−18^ and 3.29 × 10^−31^, respectively). Figure 1E demonstrates the protein expression pattern across the members of the EIF2 complex. Of note, the diagrams include the eukaryotic initiation factor 2A (EIF2A), which participates in similar biological processes as EIF2 yet is not a functional homolog of EIF2α as it functions in a codon- rather than GTP-dependent manner. The role of EIF2A in translation is limited to specific contexts, such as re-initiation, internal, and non-AUG initiation [31]. 

### 3.2. EIF2α Tissue Abundance and Phosphorylation Are Elevated in HNSCC

To validate the results in an independent cohort, tissue samples from 83 patients with HNSCC originating from the larynx, epipharynx, hypopharynx, oropharynx, and oral cavity were stained by immunohistochemistry (IHC) and evaluated for EIF2α expression. Table 1 shows the characteristics of the group studied. The neighboring non-neoplastic epithelium was used as a control (n = 55). EIF2α is present predominantly in the cytoplasm. Staining intensity is significantly stronger in cancer than in controls (*p* < 0.0001). Similarly, the total immunostaining score (TIS), which represents the proportion and intensity of stained cells, is significantly higher in tumors than in controls (*p* < 0.0001), with the median TIS value in cancer being twice that of control (Figure 2A–C).

The pathogenesis of HNSCC involves a gradual progression from dysplastic epithelium to invasive cancer, triggered and accompanied by alterations in signaling pathways. To further characterize the expression at different stages of HNSCC development, the cohort was narrowed down to samples (n = 41) where carcinoma, as well as intermediate stages of HNSCC development (i.e., dysplasia and carcinoma in situ and metastasis), could be delineated. Strikingly, the progression of the disease is mirrored by the increase in EIF2α staining intensity and TIS. The TIS score is highest in cancer metastasis (median TIS = 9), followed by invasive cancer (TIS = 8), carcinoma in situ (TIS = 7), dysplasia (TIS = 6), and lowest in neighboring normal tissue (TIS = 4) (*p* < 0.0001) (Appendix A). 34% of cancer samples feature strong EIF2α staining intensity, compared to none in the control group. Conversely, a low intensity score (IS) is seen in over 80% of controls and only 26% of cancers. Dysplasia and carcinoma in situ show intermediate staining intensity, with around 40% of samples featuring moderate IS (Figure 2D,E).

Next, we asked whether tumor cells have higher steady-state EIF2α phosphorylation levels than normal epithelium. Immunoblot analysis of freshly frozen intraoperative tissue samples (n = 16) once again confirmed higher expression of EIF2α in the tumor than in the adjacent normal epithelium (*p* = 0.0021). The abundance of phosphorylated EIF2α was also significantly elevated in cancer (*p* = 0.0125) (Figure 3, Appendix A), consistent with the hypothesis that cancers benefit from the cytoprotective aspect of increased basal EIF2α phosphorylation. The phosphorylated and unphosphorylated EIF2α ratio was higher in HNSCC, but the trend did not reach statistical significance (*p* = 0.0583).

### 3.3. Differential EIF2α Expression in HPV-Positive and Negative HNSCC

Persistent high-risk human papillomavirus (HPV) infection is a known driver of carcinogenesis, especially in the oropharynx and oral cavity. *EIF2S1* is elevated in both HPV-positive and negative HNSCC, yet the expression is higher in HPV- (n = 434) than in HPV+ cancers (n = 80; *p* = 1.36 × 10^−3^) (Appendix A). Viral proteins E6 and E7 mediate the degradation of key regulatory proteins, including TP53 and retinoblastoma (RB), and promote cell-cycle re-entry by the host cell, allowing further amplification of the viral genome [32]. To assess EIF2α expression in relation to the HPV status in our cohort, IHC for E6/E7 was performed. Eighty percent of HPV-related cancers featured moderate or strong EIF2α IS, and only 63% of HPV+ HNSCC. In the clinical setting, P16 is used as a surrogate IHC marker for transcriptionally active, high-risk HPV infection with prognostic significance. P16 positivity is reported when there is 70% or more cytoplasmic and nuclear expression with at least moderate to strong intensity. The above trend for higher EIF2α expression in HPV-HNSCC could also be seen with P16 expression (Appendix A).

### 3.4. Treatment with an EIF2α Dephosphorylation Inhibitor Decreases Cell Viability and Clonogenic Survival by Disrupting Cell Cycle Progression

A small-molecule compound, salubrinal (Sal003, S4451), inhibits dephosphorylation of EIF2α by GADD34/PP1 and CREP/PP1 complexes in a dose-dependent manner. Proteomics studies report no global changes in protein phosphorylation levels following treatment with salubrinal, and thus a remarkable substrate specificity [33]. Incubation of human HNSCC cells, FaDu and SCC4, with 10, 20, and 50 μM salubrinal increased eIF2α phosphorylation levels (Appendix A). Prolonged treatment with the inhibitor decreased cell viability starting at 24 h and even more so after 48 and 72 h. Treatment with salubrinal also drastically impaired colony-forming ability (Figure 4A–D). The decreased cell viability following treatment with salubrinal was not associated with caspase and poly(ADP-ribose) polymerase 1 (PARP) cleavage. We then set out to determine whether persistent EIF2α phosphorylation could disrupt cell cycle progression. FaDu cells were incubated with 10, 20, and 50 μM salubrinal for 36 h and stained with propidium iodide (PI) in the presence of RNAse. Cell cycle analysis with flow cytometry showed a significantly higher percentage of cells in G_0_/G_1_ in samples treated with salubrinal and fewer cells entering S and G_2_/M phases (multiple *t*-tests; *p* < 0.05) (Figure 4E,F).

Cell cycle progression requires a coordinated interaction of cyclins and cyclin-dependent kinases. At the entry point, D-type cyclins complexed with CDK4/6 phosphorylate RB1, releasing E2 transcription factor 1 (E2F1) [34]. The activated transcription factor targets multiple genes, including cyclins A and E, driving passage from the G_1_ to the S phase. Immunoblot analysis of cell cycle components revealed a decrease in cyclin D1 after treatment with salubrinal, whose abundance is highly dependent on translation rate and is rate-limiting in cells with intact RB signaling. Treatment with salubrinal also decreased phosphorylation levels of RB1 and depleted E2F1 and cyclin A. We also observed a strong induction of the CDK-inhibitor P21, which restricts S phase entry [35]. Taken together, the data suggest an impairment of the G_1_/S transition after salubrinal treatment (Figure 4G, Appendix A).

### 3.5. Salubrinal Decreases the Viability of Patient-Derived 3D Tumor Spheres (PD3DS) and Enhances the Cytotoxicity of Selected Chemotherapeutics

Patient-derived spheroids were expanded and used for chemosensitivity testing. The luminescence signal in samples treated with increasing doses of salubrinal was compared to vehicle (DMSO), and dose-response curves were generated using non-linear regression. Salubrinal decreased HNSCC spheroid viability with an IC_50_ ranging from 15.63 to 73.55 μM (Figure 5). Interestingly, the lowest dose of salubrinal was non-toxic and slightly increased PD3DS viability, consistent with the idea of ER stress alleviation.

The mainstay of therapy for inoperable HNSCC is platin-based chemotherapy combined with irradiation. The widely used chemotherapeutic TPF regimen comprises a taxane drug, a platinum compound, and 5-fluorouracil (5FU). Accordingly, we looked at FaDu viability after 48 h of co-treatment with salubrinal and paclitaxel, cisplatin, 5FU, or radiomimetic bleomycin [7]. The obtained synergy scores represent the average excess response to a given drug combination compared to the expected response assuming no interaction. Scores above 10 indicate a synergistic interaction of two agents; scores below −10 indicate antagonistic interactions; and scores between −10 and +10 indicate additive effects [29]. We observed a strong synergy between salubrinal and cisplatin (synergy score 35.02). Adding a minimally toxic dose of salubrinal (10 μM) caused a synergistic potency shift across all doses of cisplatin. The cisplatin efficacy shift is most pronounced at low doses, with relative viability inhibition increasing from 33.96% to 89.43%. Similarly, 5FU and bleomycin work synergistically with salubrinal (synergy scores of 13.77 and 11.42, respectively). The effect of paclitaxel and salubrinal is likely additive (10.21) (Figure 6A–D). Proteasome inhibitors (PI) cause cancer cell death by disrupting proteasome homeostasis. The combination of salubrinal and bortezomib is highly effective in multiple myeloma and eradicates PI-resistant cells [36,37]. We observed a synergy between salubrinal and bortezomib (score 13.13), yet the synergy score was nearly twice higher (25.07) when salubrinal was combined with MG-132, known to have a more pleiotropic mode of action. At 0.2 μM MG-132, salubrinal induced a potency shift from 6.6% inhibition to 44.1% and 52.6% when adding 10 and 50 μM, respectively (Figure 6E,F).

## 4. Discussion

Eukaryotic initiation factor 2α (EIF2α) is an integrator of environmental signals implicated in tumorigenesis. It is upregulated in many cancer types, precipitating diverse effects. A high abundance of phosphorylated EIF2α is associated with favorable prognosis in gastrointestinal carcinoma [38], breast cancer [39], and non-small-cell lung cancer [13]. On the other hand, in pancreatic cancer [40] and *PTEN*-null prostate carcinoma [41], high expression of phospho-EIF2α is associated with disease recurrence.

In this study, the *EIF2S1* gene and its encoded protein were overexpressed in head and neck squamous cell carcinoma (HNSCC) and associated with shorter overall and disease-free survival. The phosphorylated form of the protein was also overexpressed in intraoperative samples, and a trend for a higher proportion of phosphorylation in HNSCC was seen. The observation that EIF2α abundance increased as epithelial lesions progressed from dysplasia to invasive cancer holds potential clinical significance. Carcinogen exposure, such as tobacco smoke and chronic inflammation, leads to changes in the mucosa, known as field cancerization. Diverse degrees and types of genetic and epigenetic alterations render exposed mucosal areas susceptible to cancerization. As a result, multiple primary tumors are exceptionally common in HNSCC, and individual cancerous lesions are frequently surrounded by macroscopically normal but dysplastic epithelium [42]. Identifying markers that allow the delineation of malignant and premalignant lesions would limit surgical interventions to loci where they are strictly necessary. In addition, defining targets for preventive measures in high-risk groups of patients is of utmost interest.

We observed elevated *EIF2S1* expression in human papillomavirus (HPV)-positive cancers compared to the normal epithelium and even higher expression levels in HPV-negative samples. While HPV-HNSCC frequently features mut-*P53*, elevated pRB, and low P16 expression, HPV+ cancers exhibit low P53 and pRB1 levels due to proteasomal degradation and elevated P16 [43]. The underlying mechanism involves PKR sensing dsRNA and reducing global translation via EIF2α to curb viral replication. EIF2α phosphorylation depletes E6, but the remaining pool of the protein associates with GADD34/PP1 and promotes EIF2α dephosphorylation, partly restoring its own transcription. This allows HPV-infected cells to avoid apoptosis [44]. Whether pharmacologically induced phosphorylation of EIF2α can be a preventive strategy in high-risk patients with HPV is an interesting research question for future studies.

It is well documented that EIF2α phosphorylation has three possible outcomes: cytoprotection, growth arrest, and apoptosis [45]. Expression of an unphosphorylatable mutant EIF2α (in cells with low basal phosphorylation of EIF2α) triggered malignant transformation in murine and human fibroblasts. In contrast, a phosphomimetic mutant induces apoptosis [46]. A similar observation was made by Ranganathan et al. [47] in squamous carcinoma cells with low basal PERK-EIF2α signaling; reinforcing this axis resulted in G_0_/G_1_ arrest but not apoptosis. Kinases phosphorylating EIF2α have been studied extensively; both the inactivation and stimulation of their activity were considered potential strategies for anticancer therapy. Though the phosphorylation occurs at the same serine residue, the consequences are often diverse as each enzyme modifies other targets [19].

Basal levels of phospho-EIF2α mediate adaptive responses to background stress. In normal conditions, EIF2α phosphorylation is counteracted by CREP (*PPP1R15B*), a constitutively active subunit of the PP1 phosphatase complex [48]. GADD34, in contrast, is stress-inducible and specifically dephosphorylates EIF2α, allowing the translation of a subset of mRNA involved in cytoprotective stress response programs. In severe stress, cells with defective GADD34 accumulate high levels of phospho-EIF2α and are unable to induce gene expression programs facilitating survival and recovery [20]. GADD34-mediated de-repression of translation determines the availability of both long-term adaptors to stress as well as intermediate mediators required to invoke stress-response gene expression programs [20,21]. 

### 4.1. Pharmacological Modification of EIF2α Phosphorylation

Salubrinal is mechanistically different from kinases as it directly counters phosphatase activity. Physiologically, EIF2B is less abundant than EIF2α. Therefore, even subtle changes in the phosphorylation of EIF2α are impactful. While the substrate specificity of salubrinal is well established, its exact mechanism of action is not known. It most likely affects both GADD34 and CREP [33]. By inactivating GADD34/CREP, salubrinal disrupts the negative feedback loop determining survival in stress and recovery. Therefore, a knockdown of GADD34 or CREP alone is likely not directly comparable. It is known that GADD34-null mice are viable and largely unaffected [21]. On the other hand, CREP-null mice are viable but smaller and do not thrive. A simultaneous ablation of both GADD34 and CREP leads to embryonic lethality [49]. We analyzed the effects of GADD34 and CREP ablation bioinformatically using data obtained by CRISPR-Cas9 on 70 HNSCC cell lines provided by the DepMap Portal [50,51,52,53]. EIF2S1 is a common essential gene in HNSCC, as are other members of the EIF2 complex. In contrast, EIF2 kinases and phosphatases—PPP1R15A (encoding GADD34) and PPP1R15B (CREP)—are not. The dependency scores in FaDu and SCC4 cells suggest that both cell lines are more dependent on CREP (dependency scores 0.91 and 0.92, respectively) than on GADD34 (0.064 and 0.01, respectively) (Appendix A).

The concept of EIF2α hyperphosphorylation triggering cell death is well illustrated by a study on glioblastoma, in which irradiation activated PERK, triggering a cytoprotective cascade. However, irradiation combined with salubrinal enhanced cytotoxicity [54]. Previous studies have shown a moderate efficacy of salubrinal in vitro and in vivo and an enhanced efficacy in combination with other therapeutic agents. In ER+ breast cancer, phosphorylation of EIF2α by salubrinal induced ATF4 and C/EBP, triggering apoptosis, which was potentiated by 4-hydroxytamoxifen [55]. In triple-negative breast cancer salubrinal and similar in its mode of action, guanabenz [56] reduced proliferation, invasion, and motility of cancer cells. Salubrinal also reduced tumor size when injected into murine tumors [57]. Likewise, in chondrosarcoma, salubrinal was found to reduce proliferation and sensitize it to irradiation. It was also more effective in chondrosarcoma cells than in normal chondrocytes [58]. Finally, combining salubrinal with SoC agents, such as proteasome inhibitors [36,37] and doxorubicin [59], improves therapy outcomes.

### 4.2. The Effects of EIF2α Hyperphosphorylation on the Cell Cycle

Cell cycle control is frequently altered in HNSCC. Loss-of-function mutations of key tumor suppressors P53 and P16 are prevalent in HNSCC and allow cells to bypass the G_1_/S checkpoint [60]. P16 interrupts the formation of cyclin D/CDK complexes. Its frequent inactivation, coupled with cyclin D1 amplifications, present in up to 28% of HNSCC, facilitates S phase entry and replication. Moreover, the inhibitory effect of P53 on the cell cycle resulting from P21 activation is often abolished [8]. In the current report, prolonged treatment with salubrinal induced G_1_/S arrest. According to the literature, endoplasmic reticulum (ER) stress can be associated with cell cycle arrest at G_1_/S or G_2_/M checkpoints. G_2_/M arrest can be attributed to the depletion of cyclin B1 [61], which was not confirmed in this study. Instead, EIF2α phosphorylation downregulated the synthesis of cyclins D1 and A. Cyclin D1 is an unstable protein; its abundance directly depends on translation rate [62].

Our observations are also in agreement with previous studies showing an induction of P21, a CDK inhibitor and tumor suppressor, via an EIF2α-dependent mechanism [63]. Recently, Darini et al. [64] demonstrated that salubrinal potentiates the antitumor activity of trastuzumab in HER2+ breast cancer xenografts by increasing P21 and JNK1/2 activity, thus providing a robust experimental rationale for the therapeutic use of salubrinal in breast and gastric cancers.

A novel aspect of this study is the depletion of E2F1 following EIF2α phosphorylation. E2F1, though a tumor suppressor in normal cells, in HNSCC promotes proliferation, invasion, suppresses squamous differentiation, and inhibits apoptosis [65]. It is elevated in aggressive and metastatic cancers and is a marker of poor prognosis [66]. CDK inhibitors (palbociclib, ribociclib) indirectly target E2F1. Results of clinical trials in HNSCC are still emerging but appear promising in combination with cisplatin, 5-fluorouracil (5FU), and docetaxel [7]. It would be interesting to test whether additional pharmacological phosphorylation of EIF2α with salubrinal is beneficial. Finally, E2F1 (reduced in response to salubrinal) is implicated in therapy resistance to many genotoxic agents such as cisplatin, doxorubicin, and etoposide by, among other mechanisms, inducing efflux transporters [67,68]. 

### 4.3. Salubrinal-Induced Sensitization to Chemotherapeutics

Intrinsic and acquired drug resistance is a major obstacle in HNSCC therapy; the response rates for cisplatin, taxanes, and 5FU range between 13 and 40%. At the same time, these drugs have significant systemic toxicity [69]. We conducted a concentration-weighted analysis to determine synergies occurring at low drug doses to minimize off-target toxicity and observed a strong synergy between salubrinal and cisplatin. Salubrinal also worked synergistically with other genotoxic drugs (5FU and bleomycin). There are several possible mechanisms contributing to this effect. During apoptosis, the dominant cytotoxic mechanism of anticancer drugs, caspases cleave EIF2α. (1) As the ternary complex becomes less abundant, pharmacological interventions on a smaller pool of the regulatory subunit become more effective [70]. (2) PKR senses DNA damage caused by genotoxic agents and *pre*-phosphorylates EIF2α. (3) It is also possible that salubrinal affects the cellular metabolism of other chemicals.

Salubrinal induces translocation of calreticulin to the cell surface, which stimulates the antitumor immune response in the presence of genotoxic agents [71]. Treatment with salubrinal combined with mitomycin or etoposide causes tumor regression in immunocompetent mice but not Nu/Nu [72]. While cisplatin alone does not induce immune cell death (ICD) [73], it is conceivable that the addition of salubrinal would escalate the cytotoxic effect in vivo even more due to the immunogenic component. Anticancer therapy can induce clinically relevant immunogenic antitumor responses, and harnessing them with small-molecule ICD inducers, auxiliary to chemotherapy, is currently being explored [74]. Bleomycin is an ICD inducer, as is radiation therapy. While the immunogenic properties of 5FU are debated, it likely modulates the tumor microenvironment rather than induces immunogenic cell death [73]. The main cytotoxic effects result from its pleiotropic genotoxicity, but 5FU also affects tumor cells by generating mitochondrial ROS, inhibiting angiogenesis, and influencing the cell cycle. Similar to salubrinal, 5FU was found to deplete cyclin D1, induce P21, and lead to G_1_/S arrest [75,76].

The addition of paclitaxel, a microtubule-stabilizing drug, to cisplatin and 5FU was shown to further improve therapeutic efficacy in HNSCC, establishing the TPF regimen [77]. Paclitaxel typically induces a strong stress response and phosphorylates eIF2α via PERK and GCN2. Therefore, it was surprising that the combination with salubrinal did not lead to a significant potency shift. However, the functional relationship between the microtubule (MT) cytoskeleton and eIF2α signaling is complex. First, eIF2α phosphorylation is necessary to gear translation towards the production of MT components, then dephosphorylation to initiate MT organization and clear misfolded protein aggregates [78].

Lastly, we observed a strong synergy between salubrinal and proteasome inhibitors (PI), previously described in hematological malignancies [36,37]. Bortezomib selectively inhibits the chymotrypsin-like activity of the 26S proteasome complex. It is approved for clinical use in multiple myeloma and mantle cell lymphoma, malignancies highly dependent on proteasomal degradation of toxic protein aggregates [36]. Bortezomib was evaluated with SoC therapy for HNSCC in phase II clinical trials but failed to improve overall and progression-free survival significantly [79]. MG-132, used for research purposes, is less specific and has off-target effects on protein synthesis, cell cycle, and ubiquitination. The potency shift upon co-treatment with MG-132 and salubrinal is likely a result of the less well-defined effects of the inhibitor.

## 5. Conclusions

The expression of EIF2 is elevated in HNSCC at both RNA and protein levels. *EIF2S1* is associated with shorter overall and disease-free survival and clinicopathological features indicating disease severity and aggressiveness. The tissue abundance of EIF2α increases across premalignant and malignant lesions, mirroring the progression to malignancy. Consistent with its role in response to stress signals, the steady-state phosphorylation level in HNSCC is higher than in normal epithelium. EIF2α is also more abundant in non-viral HNSCC, associated with carcinogen exposure. Our data demonstrates that pharmacological hyperphosphorylation of EIF2α with the small-molecule inhibitor salubrinal decreases cell viability in adherent cell cultures as well as in patient-derived 3D tumor spheres. This effect is mediated by an impairment of cell cycle progression past the G_1_/S checkpoint. Salubrinal also works synergistically with genotoxic drugs used in HNSCC management: cisplatin and 5-fluorouracil, bleomycin, and proteasome inhibitors. Future studies should evaluate salubrinal in combination with genotoxic agents in an in vivo model of HNSCC. Existing data suggests that such a combination may prove even more beneficial due to the induction of an antitumor immune response. Pharmacological hyperphosphorylation could be a potential strategy for HNSCC prevention in selected high-risk patients.

## 6. Patents

Haybaeck J., Cyran A. M., Arens C., and Naumann M., Eukaryotic translation initiation factors (EIFs) as novel biomarkers in head and neck squamous cell carcinoma (HNSCC), EP 3 725 898 B1, 9 February 2022.

## Figures and Tables

**Figure 1 cancers-15-05350-f001:**
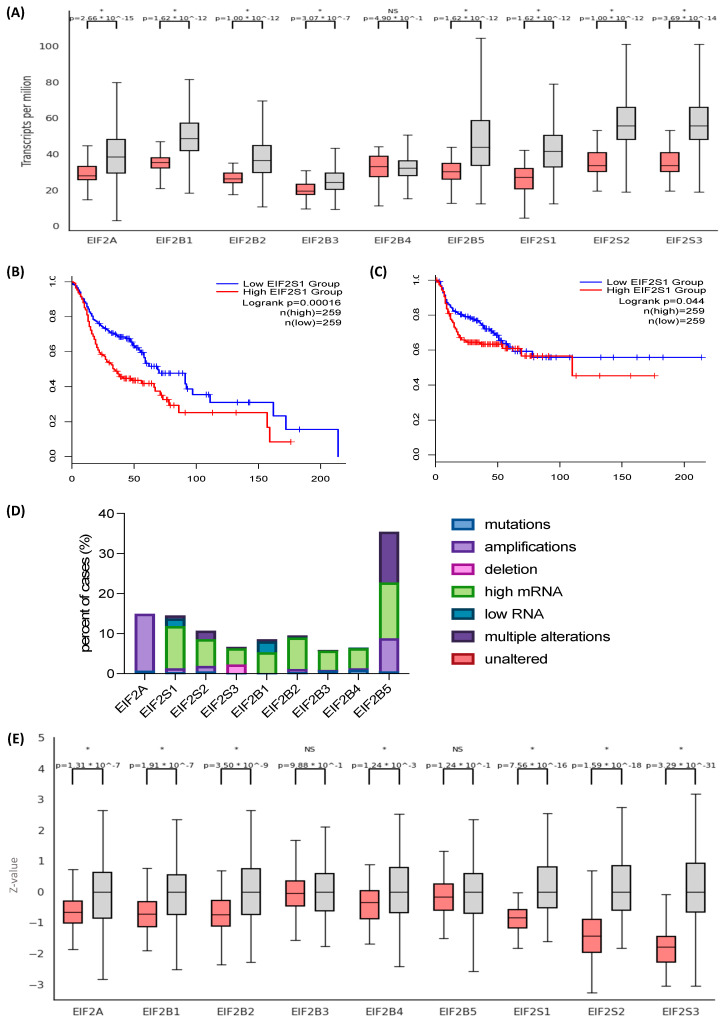
*EIF2S1* and its protein product are overexpressed in head and neck squamous cell carcinoma (HNSCC). (**A**) *EIF2S1* transcript expression is significantly higher in HNSCC (n = 520) than in normal epithelium (n = 44) (*p*-value = 1.62 × 10^−12^). Red and gray boxes represent normal and neoplastic tissue, respectively. (**B**) High *EIF2S1* expression correlates negatively with overall survival (OS) and (**C**) disease-free survival (DFS) in a cohort of 518 patients with HNSCC. Patients were divided into low- and high-expressors, with a cut-off point at the median. (**D**) Genetic alterations driving the expression of EIF2 components. *EIF2S1* and genes encoding other eIF2 subunits are rarely mutated, yet their expression is frequently deregulated as a result of amplifications or changes in upstream signaling. (**E**) The protein abundance of eIF2 complex members is significantly higher in HNSCC (n = 108) than in non-neoplastic tissues (n = 71). Red and gray boxes represent normal and neoplastic tissue, respectively. Statistical significance level *p* < 0.05 (*); not significant (ns).

**Figure 2 cancers-15-05350-f002:**
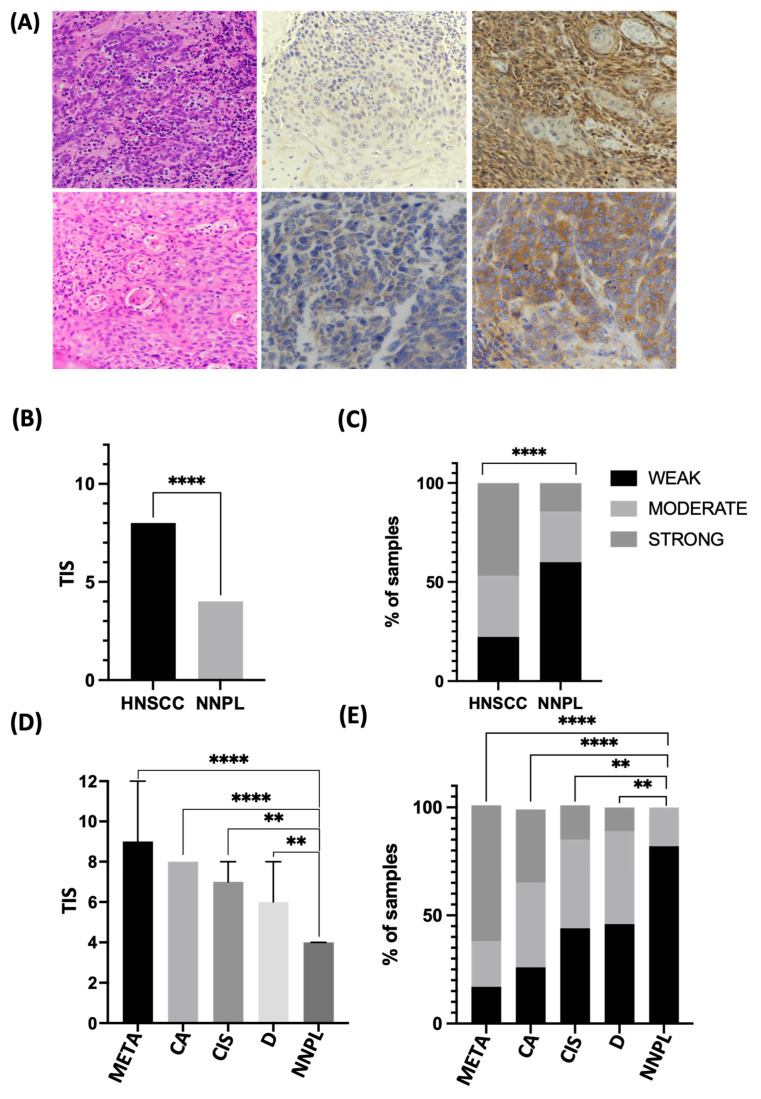
eIF2α is overexpressed in head and neck squamous cell carcinoma (HNSCC). (**A**) Representative photomicrographs of HNSCC slides showing H&E and immunohistochemical stainings. Top row (20×) in respective order: H&E, low eIF2α expression (total immunostaining score = 2), and high eIF2α expression (TIS = 12). Bottom row (40×): H&E, low and high eIF2α expression. (**B**) TIS is significantly higher than HNSCC in adjacent non-neoplastic tissue (NNPL) (Mann-Whitney U test; *p* < 0.0001). (**C**) eIF2α staining intensity is significantly stronger in HNSCC than in adjacent NNPL (X^2^ test; *p* < 0.0001). (**D**) TIS for eIF2α staining is highest in cancer metastasis, followed by HNSCC primum and precancerous lesions (X^2^ test; * *p* < 0.05; ** *p* < 0.01; *** *p* < 0.001; **** *p* < 0.0001). (**E**) eIF2α staining intensity differs across stages of HNSCC development, with the highest IS observed in cancer metastasis and the lowest in tumor-adjacent non-neoplastic tissue (X^2^ test; *p* < 0.0001).

**Figure 3 cancers-15-05350-f003:**
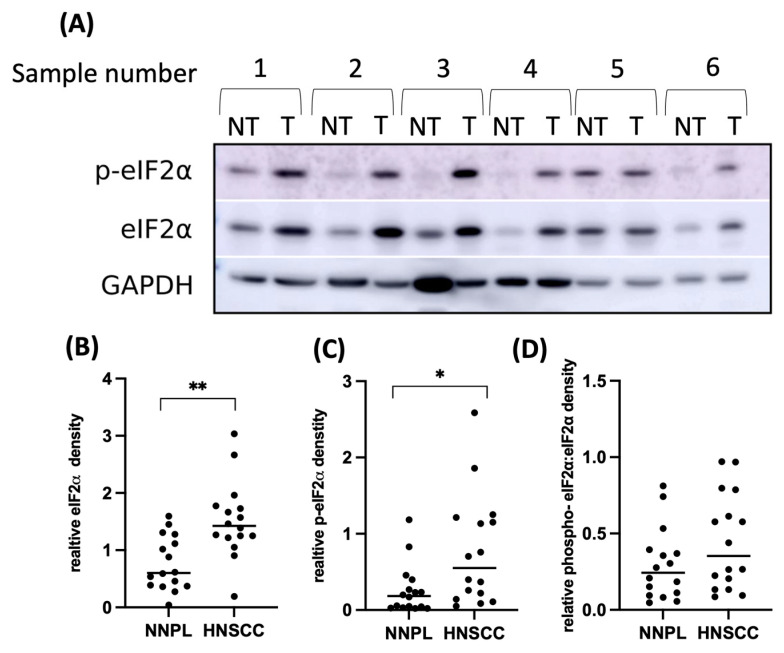
eIF2α abundance and phosphorylation are significantly increased in head and neck squamous cell carcinoma (HNSCC) in comparison to adjacent normal epithelium. (**A**) Examples of immunoblots from intraoperative tissue samples from six patients with HNSCC. For each patient, non-tumorous tissue (NT) and tumor (T) are shown. (**B**,**C**) Densitometric analyses of eIF2α and phospho-eIF2α protein expression (n = 16; *p* = 0.0021 and *p* = 0.0125, respectively; Wilcoxon test; * *p* < 0.05; ** *p* < 0.01). The intensity of the signal was normalized to GAPDH. (**D**) A trend for a higher ratio of phosphorylated eIF2α: total eIF2α is higher in HNSCC than in non-neoplastic lesions (NNPLs) (*p* = 0.0583; Wilcoxon test). Original uncropped immunoblot images and densitometric analysis are shown in Appendix A.

**Figure 4 cancers-15-05350-f004:**
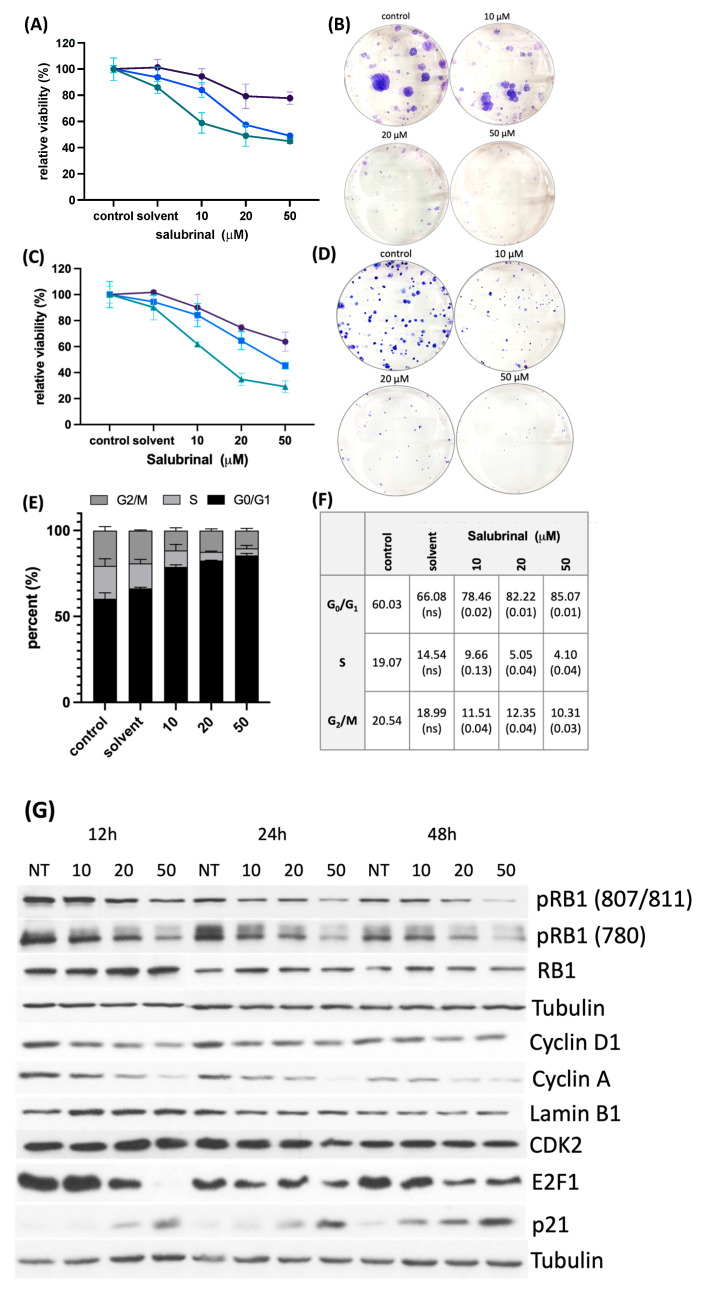
Treatment with eIF2α dephosphorylation inhibitor decreases head and neck squamous cell carcinoma (HNSCC) cell viability and clonogenic survival. (**A**,**C**) SCC4 and FaDu cells were treated with increasing concentrations of salubrinal (10, 20, and 50 μM). Viability was measured after 24 h (purple line), 48 h (blue line), and 72 h (green line) of continuous incubation. (**B**,**D**) Increasing concentrations of salubrinal markedly decreased clonogenic survival. (**E**) Treatment with salubrinal resulted in changes in cell cycle distribution in FaDu cells, with an increasing number of cells arrested in G_0_/G_1_ and fewer cells entering S and G_2_/M phases (*t*-test, *p* < 0.05). (**F**) Table showing percentages of events in each phase of the cell cycle and statistical significance in comparison to the control (*p*-value obtained with a *t*-test; not significant (ns)). (**G**) Salubrinal disrupts the progression of the cell cycle in FaDu cells. Treatment with salubrinal leads to a decrease in RB1 phosphorylation at Ser807/811 and Ser780, depletion of cyclin D1, cyclin A, and E2F1, and an induction of P21. Original uncropped immunoblot images and densitometric analysis are shown in Appendix A.

**Figure 5 cancers-15-05350-f005:**
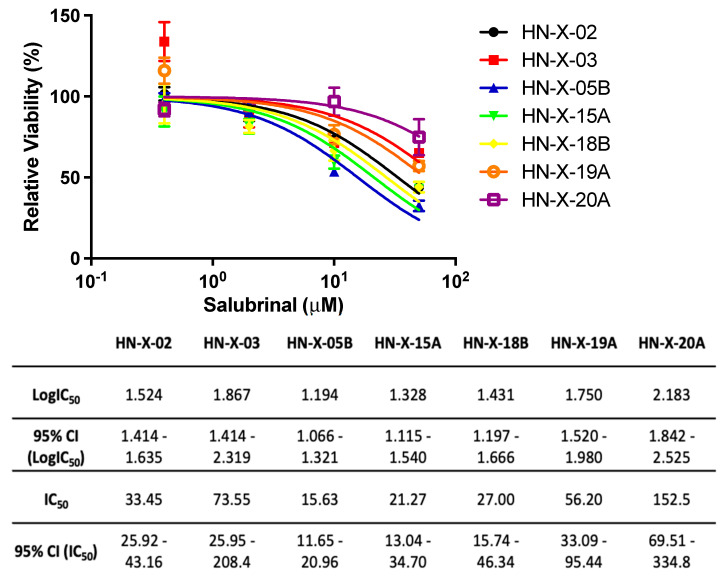
Salubrinal decreases the viability of patient-derived 3D tumor spheres (PD3DS). PD3DS were cultured on 384-well plates in a mixture of Matrigel and medium in the presence of increasing salubrinal concentrations (0.4, 2, 10, and 50 μM) for 4 days, followed by an ATP-based viability assay. Dose-response curves were obtained from four biological replicates using non-linear regression. The table represents IC_50_ and LogIC_50_ for each sample with 95% confidence intervals (CI).

**Figure 6 cancers-15-05350-f006:**
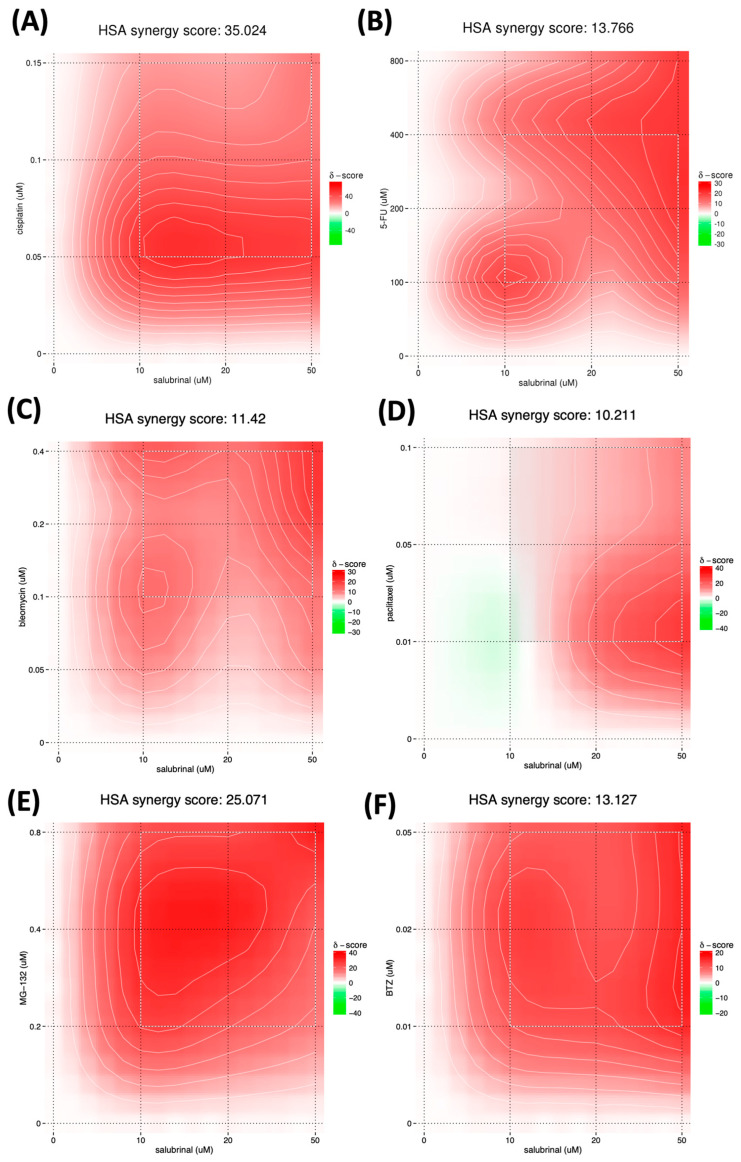
Salubrinal synergizes with standard-of-care chemotherapeutics in head and neck squamous cell carcinoma (HNSCC) in vitro. (**A**–**D**) Synergy scores and heatmaps were obtained after 48h of co-treatment with salubrinal and standard-of-care chemotherapeutics. Salubrinal has a synergistic effect with cisplatin, 5-fluorouracil, and bleomycin and an additive effect in combination with paclitaxel. Salubrinal synergizes with proteasome inhibitor (PI) bortezomib (**E**) and MG-132 (**F**) in HNSCC.

**Table 1 cancers-15-05350-t001:** Characteristics of the patient cohort used for immunohistochemical (IHC) assessment of eIF2α protein expression in head and neck squamous cell carcinoma (HNSCC).

Patient Characteristics		Number of Cases (%)
Sex	Male	73 (88%)
	Female	10 (12%)
	Total	83 (100%)
Tumor localization	Larynx	57 (69.7%)
	Pharynx	13 (15.7%)
	Tongue/tongue base	6 (7.2%)
	Tonsil	2 (2.4%)
	Other	5 (6%)
Metastasis	Localized disease	47 (56.6%)
	Nodal metastasis	16 (19.3%)
	Distant metastasis	15 (18.1%)
	N/D	5 (6.0%)
HPV status	Positive	11 (13.2%)
	Negative	66 (79.5%)
	N/D	6 (7.2%)
P16 status	Positive	20 (24.1%)
	Negative	56 (67.5%)
	N/D	7 (8.4%)

## Data Availability

Additional data generated in this project are available in the Appendix A provided and, upon reasonable request, via the correspondence author.

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
