# Peer review of "Inhibition of EIF2α Dephosphorylation Decreases Cell Viability and Synergizes with Standard-of-Care Chemotherapeutics in Head and Neck Squamous Cell Carcinoma"

_cancers, 2023, doi:10.3390/cancers15225350_

Round 1
Reviewer 1 Report
Comments and Suggestions for Authors
This study demonstrated that EIF2α gene transcripts, protein level and phosphorylation level are elevated in head and neck squamous cell carcinoma (HNSCC), and pharmacological inhibition of EIF2α dephosphorylation by Salubrinal decreases HNSCC cell viability and acts synergistically with standard-of-care drugs. Overall, the results appear interesting, especially it is closely related to clinical therapeutics. the manuscript can be accepted for publication if a minor revision is well-addressed. Below are some specific comments:
1. In Figure 4, SCC4 and FaDu cells were used for Salubrinal-treated cell viability assays, it is needed to confirm the level changes of phos-EIF2α and EIF2α protein in immunoblots using these two cell lines upon Salubrinal treatment.
2. Besides small-molecule inhibition, it’s better to use another technique, such as knockdown or knockout of GADD34/PP1 and CREP/PP1 complex to test the EIF2α dephosphorylation hypothesis.
Author Response
Research Article: Inhibition of EIF2α dephosphorylation decreases cell viability and synergizes with standard-of-care chemotherapeutics in head and neck squamous cell carcinoma
Response to Reviewer's Comments:
On behalf of all authors, I want to thank you for taking the time to review our manuscript. Your contribution to the scientific community is appreciated.
Point 1:
Western blot confirmation of increased eIF2alfa phosphorylation levels ( and total eIF2alfa) upon treatment with salubrinal for FaDu and SCC4 cells is now provided in Supplementary Figure 8. The newly added results are referenced in the main text in lines 343- 346 (marked in red). Original, uncropped western blots and comprehensive densitometric analysis is provided in Supplementary material 10.
Point 2:
We pursued the question posed by the reviewer by means of a bioinformatic analysis. Publicly available data (DepMap.org) from CRISPR-Cas9 depletion assays on 70 HNSCC cell lines were analyzed and presented as two measures:
- The CRISPR Gene Effect describes the perturbation effects of a given gene in cells. Scores close to 0 indicate non-essential genes, while negative scores indicate a higher likelihood that the gene of interest is essential.
- The Dependency Score of 0 indicates that a gene is not essential; score of 1 corresponds to the median of all pan-essential genes.
Accordingly, EIF2S1 is a common essential gene in HNSCC, while GADD34 (PPP1R15A) and CREP (PPP1R15B) are not. Across 70 HNSCC cell lines tested, the effect of PPP1R15A (GADD34) knockdown is close to zero. Similarly, the dependency score is close to zero. The same is true for EIF2 kinases, which are included in the diagram for demonstration. Knockdown of the constitutive phosphatase subunit PPP1R15B (CREP) is more consequential for HNSCC cells, but its effects vary between cell lines. Dependency Scores in FaDu and SCC4 cells suggest that both cell lines are more reliant on CREP (Dependency score 0.91 and 0.92 respectively) than on GADD34 (0.064 and 0.01 respectively).
While the substrate specificity of salubrinal is well established, its exact mechanism of action is not known. It most likely affects both GADD34 and CREP (Boyce et al. 2009, Science). Therefore knockdown of GADD34 or CREP alone is likely not directly comparable. It is known that GADD34 null mice are viable and largely unaffected (Kojima 2003; Patterson 2006). CREP null mice are viable but smaller and do not thrive (Harding 2009). However, a simultaneous ablation of both GADD34 and CREP leads to embryonic lethality (Harding 2009 PNAS).
Lastly, the focus of our study was on the therapeutic potential of eIF2alfa signaling modulation, hence the use of a small-molecule inhibitor. In contrast, a gene knockdown or knockout could have more pleiotropic effects on cell biology.
The results of the bioinformatic analysis are provided as Supplementary Figure 9. The figure is referenced in the main text and a summary of the points made above is included in the main text as well (lines 485- 497, all changes are marked in red).
Reviewer 2 Report
Comments and Suggestions for Authors
Dear Editors,
Authors described the expression pattern of EIF2S1, encoding the regulatory subunit of EIF2 in HNSCC. They determined its protein abundance in carcinoma and premalignant epithelial lesions and related the expression to clinical and histopathological traits. They then evaluated the effects of pharmacological targeting of EIF2 by a small-molecule inhibitor to determine its therapeutic potential in HNSCC alone and in combination with standard-of-care drugs. My comments are listed below;
-Abstract is ok
-Introduction is poor and I suggest to use the below references to improve this section about HNSCC and ....;
*Mosaddad SA, P Mahootchi, Z Rastegar, B Abbasi, M Alam, K Abbasi, S Fani-Hanifeh, S Amookhteh, S Sadeghi, RS Soufdoost, M Yazdanparast, A Heboyan, H Tebyaniyan and GVO Fernandes: Photodynamic Therapy in Oral Cancer: A Narrative Review. Photobiomodul Photomed Laser Surg,2023. DOI: 10.1089/photob.2023.0030
*Khayatan D, A Hussain and H Tebyaniyan: Exploring animal models in oral cancer research and clinical intervention: A critical review. Vet. Med. Sci,2023. DOI: https://doi.org/10.1002/vms3.1161
*Mosaddad SA, K Beigi, T Doroodizadeh, M Haghnegahdar, F Golfeshan, R Ranjbar and H Tebyanian: Therapeutic applications of herbal/synthetic/bio-drug in oral cancer: An update. Eur J Pharmacol 890: 173657,2021. DOI: 10.1016/j.ejphar.2020.173657
*Hajmohammadi E, T Molaei, SH Mowlaei, M Alam, K Abbasi, D Khayatan, M Rahbar and H Tebyanian: Sonodynamic therapy and common head and neck cancers: in vitro and in vivo studies. Eur Rev Med Pharmacol Sci 25(16): 5113-5121,2021. DOI: 10.26355/eurrev_202108_26522
-Method and material are ok
-Results section is ok
-Discussion section is poor and i suggest authors to use the above mentioned references.
Best,
Author Response
Research Article: Inhibition of EIF2α dephosphorylation decreases cell viability and synergizes with standard-of-care chemotherapeutics in head and neck squamous cell carcinoma
Response to Reviewer's Comments:
On behalf of all authors, I want to thank you for taking the time to review our manuscript. Your contribution to the scientific community is appreciated.
As requested, we included the suggested literature sources in the article in question (all changes made to the manuscript are marked in red):
0) Abstract: no changes
1) Introduction: In lines 72- 74 we provide a more comprehensive list of existing and experimental therapeutic options for HNSCC, including the following literature sources:
*Hajmohammadi E, T Molaei, SH Mowlaei, M Alam, K Abbasi, D Khayatan, M Rahbar and H Tebyanian: Sonodynamic therapy and common head and neck cancers: in vitro and in vivo studies. Eur Rev Med Pharmacol Sci 25(16): 5113-5121,2021. DOI: 10.26355/eurrev_202108_26522
*Mosaddad SA, P Mahootchi, Z Rastegar, B Abbasi, M Alam, K Abbasi, S Fani-Hanifeh, S Amookhteh, S Sadeghi, RS Soufdoost, M Yazdanparast, A Heboyan, H Tebyaniyan and GVO Fernandes: Photodynamic Therapy in Oral Cancer: A Narrative Review. Photobiomodul Photomed Laser Surg,2023. DOI: 10.1089/photob.2023.0030
*Li Q, Tie Y, Alu A, Ma X, Shi H. Targeted therapy for head and neck cancer: signaling pathways and clinical studies. Signal Transduct Target Ther. 2023 Jan 16;8(1):31. doi: 10.1038/s41392-022-01297-0. PMID: 36646686; PMCID: PMC9842704.
2) Materials and Methods: no changes
3) Results: no changes
4) Discussion: In lines 548- 555 we discuss in greater detail the mechanism of action of the microtubule-stabilizing drug paclitaxel and possible interactions with eIF2alfa signaling. We provide literature references discussing the use of paclitaxel in HNSCC:
*Mosaddad SA, K Beigi, T Doroodizadeh, M Haghnegahdar, F Golfeshan, R Ranjbar and H Tebyanian: Therapeutic applications of herbal/synthetic/bio-drug in oral cancer: An update. Eur J Pharmacol 890: 173657,2021. DOI: 10.1016/j.ejphar.2020.173657
as well as compelling evidence on the interplay between the eIF2alfa stress signaling and microtubule cytoskeleton function:
*Hurwitz B, Guzzi N, Gola A, Fiore VF, Sendoel A, Nikolova M, Barrows D, Carroll TS, Pasolli HA, Fuchs E. The integrated stress response remodels the microtubule-organizing center to clear unfolded proteins following proteotoxic stress. Elife. 2022 Jun 27;11:e77780. doi: 10.7554/eLife.77780. PMID: 35758650; PMCID: PMC9299849.
Round 2
Reviewer 2 Report
Comments and Suggestions for Authors
Dear,
The revised version is acceptable.
Best,